# Deaf Children Need Rich Language Input from the Start: Support in Advising Parents

**DOI:** 10.3390/children9111609

**Published:** 2022-10-22

**Authors:** Tom Humphries, Gaurav Mathur, Donna Jo Napoli, Carol Padden, Christian Rathmann

**Affiliations:** 1Department of Communication, University of California at San Diego, La Jolla, CA 92093, USA; 2Department of Linguistics, Gallaudet University, Washington, DC 20002, USA; 3Department of Linguistics, Swarthmore College, Swarthmore, PA 19081, USA; 4Division of Social Sciences, Department of Communication and Dean, University of California at San Diego, La Jolla, CA 92093, USA; 5Department of Deaf Studies and Sign Language Interpreting, Humboldt-Universität zu Berlin, 10019 Berlin, Germany

**Keywords:** deaf children, sign language, bimodal-bilingual childrearing, bilingual bimodal education, individual and family well-being

## Abstract

Bilingual bimodalism is a great benefit to deaf children at home and in schooling. Deaf signing children perform better overall than non-signing deaf children, regardless of whether they use a cochlear implant. Raising a deaf child in a speech-only environment can carry cognitive and psycho-social risks that may have lifelong adverse effects. For children born deaf, or who become deaf in early childhood, we recommend comprehensible multimodal language exposure and engagement in joint activity with parents and friends to assure age-appropriate first-language acquisition. Accessible visual language input should begin as close to birth as possible. Hearing parents will need timely and extensive support; thus, we propose that, upon the birth of a deaf child and through the preschool years, among other things, the family needs an adult deaf presence in the home for several hours every day to be a linguistic model, to guide the family in taking sign language lessons, to show the family how to make spoken language accessible to their deaf child, and to be an encouraging liaison to deaf communities. While such a support program will be complicated and challenging to implement, it is far less costly than the harm of linguistic deprivation.

## 1. Introduction

While an emphasis on early language acquisition (from birth and in the first years of life) is important advice to parents, we recognize that providing a deaf child with adequate access to language remains challenging both for parents and the professionals advising them. (We use *deaf* to include all levels of hearing loss affecting linguistic input.) We argue that children who are born deaf or become deaf in early childhood need multimodal and comprehensible language exposure and engagement in joint activity with parents, caretakers, siblings, and others in their social world. For ease of exposition, we use the cover term *parents* to include all these relevant parties. In addition, we argue that choosing speech to the exclusion of another modality is a false choice that runs a greater risk of delayed development and /or language deprivation for the deaf child. Sign language and spoken language can be used side by side without compromising each other to greatly reduce the risk of harm to deaf children. There is no simple way to compare research regarding deaf children’s linguistic and cognitive development with regard to sign language versus spoken language since studies vary in how they lump together different language environments and since studies vary in how they assess linguistic and cognitive development. To make our arguments, then, we first outline the risks that face children who are deaf in early childhood. We discuss the complexities of the developmental process when a deaf child is born into a hearing family that does not have a history of sign language use or awareness. Then, we argue that children who are deaf in early childhood need to be immersed in a multimodal and multilingual environment, within which they are exposed to sign language with the frequency and richness that assures age-appropriate first language acquisition. Even when most parents of deaf newborns do not know sign language, achieving sufficient language acquisition begins at home and continues in school.

At the end, we refer the reader to Humphries and colleagues [1,2], where common questions asked by parents of deaf children are answered in a straightforward way with the best available evidence. The questions are listed in Appendix A for the reader’s reference.

## 2. Risks Facing Deaf Children

Children exposed to accessible language in early childhood will acquire it naturally. Early, multimodal, and accessible language exposure is of crucial importance: infancy is a sensitive period in the acquisition of language [3,4,5]. Delays in exposing children to accessible language carry cognitive and psycho-social risks [6,7,8]. Children with language delay, regardless of the cause of that delay, are too often “on a trajectory of academic failure and social difficulties” [9] (p. 120). This stands to reason: our ability to read and write depends on linguistic competence, and a great part of our ability to interact with other human beings depends on our ability to understand others and to be understood by others. Late learning of a first language affects the architecture of the brain [10,11,12,13,14,15], which may be (part of) why acquisition is so difficult for deaf late learners of sign languages who do not already have firm competence in a spoken language. Word order in one-clause sentences, for example, seems possible to be learned later [16] but word order in complex sentences seems less readily learned [17]. Even those with severe cognitive deficits, such as Williams Syndrome [18] benefit from early language exposure and multimodality.

Deaf adults face an increased risk of chronic health difficulties that correlate with adverse childhood communication experiences [19]. They are more likely to be under-educated [20], be un(der)employed [21,22], be abused emotionally and physically [23,24,25,26,27,28], experience food insecurity [29], experience injustices at the hand of the criminal justice system [30,31], and experience poor health outcomes [32,33,34] partly due to limited access to or inappropriate health services [35,36,37,38].

Scholars and many medical professionals know the risks of late acquisition of a language. Yet many deaf children are still not receiving adequate exposure to accessible language early enough [39]. Rather, families that seek help and sometimes want a “cure” for their child’s deafness, often are given advice that is one-dimensional, i.e., unimodal, when what the child needs is a multimodal multilingual environment.

Sign languages are natural human languages, just as spoken languages are. For many arguments and citations to that effect, we refer to the American Speech–Language–Hearing Association [40,41]. Sign languages support analogical-reasoning abilities [42], executive function [43], non-verbal working memory [44], and the wide range of cognitive abilities that spoken languages support [45]; the human need for language is satisfied equally by sign languages and spoken languages [46,47]. However, there is a crucial difference regarding accessibility: sign languages are immediately accessible to all children, hearing and deaf, including deaf/blind [48,49], while spoken languages might be fully accessible only to some deaf children and only some of the time.

For the past two decades, the Joint Committee on Infant Hearing in the United States has been recommending that deaf children’s language development be assessed routinely [50,51], but the recommendation is not widely followed, most likely because of the difficulties in properly assessing this development, given that deaf children are a heterogenous group with respect to their language access profile, where both quantity and quality of the accessible language matters [52,53]. As a result of this lack of monitoring, delays in making strides in language development might not be detected until the child is seriously behind. It is common for parents to persist with speech until it becomes undeniable that the child is not developing as hoped. At that point, they finally turn to sign language for reliable communication (see Pfister [54] for a description in Mexico City; much of the experience described is common across North America) but by then, unfortunately, significant time has been lost for the child’s cognitive development [55,56]. Children with typical language development make huge developmental strides in language and cognition in short periods of time, such that a few weeks in early childhood is significant in a child’s life. If it is not absolutely clear that a deaf child is making timely developmental strides and that there is strong, direct two-way communication between children and caregivers, no parent/caregiver should wait [19].

Sometimes parents request suboptimal treatment for their children—perhaps from denial, perhaps from previous unhelpful advice, for financial reasons, or for religious reasons—and medical professionals find themselves having to decide whether parents’ requests are in the best interest of the deaf child. In a study of such decisions in the United Kingdom, doctors were less likely to agree to the suboptimal treatment if it increased the risk of death or the risk of long-term disability than if it increased the risk of pain [57]. Similar findings (with even less willingness to allow parents to make decisions that risked harm to their children) appear in a study of American oncologists [58]. Medical professionals must transcend their own biases, and sometimes their past training, to advise parents not to wait when there is little or no evidence of the normative leaps and bounds in the child’s early linguistic and cognitive development.

The health of the overwhelming majority of deaf children is at stake in this discussion [59,60,61]. While deaf babies born to signing deaf parents, generally, are exposed to accessible language from day one; 96% of deaf babies are born to non-signing hearing parents [62], and it is those children who are most at risk of delayed language development. Many medical professionals advise parents to have their child undergo surgery for a cochlear implant (CI) and, for hearing parents, to raise the child with speech and hearing only [63]. This advice has led parents to trust that a CI will, essentially, make their child “hearing”, often giving them an unwarranted sense of security in their decision. However, success with CI is enormously variable: a CI alone often cannot be relied upon to provide the kind of language access necessary for healthy language development [60,64]. Many children with a CI or bilateral CIs do not have sufficient access to spoken language input for timely language and cognitive development, even after years of intensive aural rehabilitation (auditory training and speech exercises) with family and speech therapists [65,66,67,68,69,70,71]. Perhaps most important of all, we lack ways to accurately predict the likelihood of the benefit of a CI for individual deaf children [2,72,73,74].

The ethical action for medical professionals is to avoid assuring parents that a CI is a clear path to hearing and language. Three kinds of evidence to this effect are readily available to scholars and medical professionals. First, deaf children lag behind their hearing peers in language development even with a CI or bilateral CIs [75,76], and even when they have been implanted at a very early age [77]. Delays are evident in their vocabulary, reading, and writing abilities [75,78,79,80,81,82]. Children with CIs exhibit weaknesses in complex, high-order language processing across the board [83]. While some implanted children demonstrate semantic processing of vocabulary after 12 months of CI use, others fail to show improved processing after two years of CI use. Since that time period goes past the sensitive period for language acquisition for the latter group, it is no surprise that that group had poor language outcomes overall [84].

Second, if a medical device is providing what is needed—i.e., if it is effective—then deaf people themselves will be incentivized to use it. However, CI use is not uniform, implicitly calling into question its efficacy. Studies have varied on continued CI use 10 years after implantation: from 63% to 87.8% [85,86,87,88,89]. There is also considerable disagreement among members of deaf communities as to whether being implanted in childhood improves their daily lives or not [90], where many point to limitations in auditory perception as a reason for discontinuing CI use [91]. Variation in efficacy for language acquisition is not the same as efficacy for access to sound, further complicating the benefits of CI use.

Third, unlike normal hearing, CI input is not “on” all the time, and children often use CIs inconsistently. They are not or cannot be used during certain play or athletic activities. On a daily basis, and in entirely ordinary circumstances, background noise in the environment is a factor, and that is when CIs make the job of distinguishing language exceedingly difficult [92]. Two critically important environments are school and the family dinner table. If such environments do not afford intact and visual language input via sign language, the child has reduced opportunities for incidental or contextual learning [45,93].

## 3. Sign Language Exposure Ensures Accessibility

Here we address questions related to the benefit of sign languages for a deaf child. We distinguish natural sign languages, used by deaf communities, from manually coded versions of spoken languages, such as ”Signed Exact English” (SEE), and other manual codes for spoken language. We know of no evidence that using such codes exclusively offers a viable or sufficient first language foundation (cf. [94]).

Sometimes the rationale for denying a deaf child a sign language is based on the assumption that learning a sign language would compete for the child’s attention and possibly change the brain in such a way as to interfere with learning a spoken language. These assumptions are false: no such interference in brain behavior or cognition occurs [95,96,97]. Indeed, sign languages and spoken languages both are robustly left hemisphere-lateralized with respect to production, regardless of the difference in the modality of language input [98], and sign languages and spoken languages are cortically organized in terms of specialization of the dominant perisylvian system [99]. In particular, regardless of whether a child signs or speaks, their oculomotor behavior during reading is the same [100]. A minimum expectation, then, should be that learning to sign would not interfere with spoken language abilities at all, no matter how early bimodal-bilingualism is introduced [101,102,103,104].

On the contrary, signing benefits the cognitive abilities of the deaf child that is also learning a spoken language, just as bilingualism between spoken languages benefits cognitive abilities [105], particularly bilingualism in the early years of life [106]. Early bimodal-bilingualism affects neural activation in a way that may well enhance linguistic and cognitive processing [107]. In fact, regardless of the use of a CI, sign language proficiency positively correlates to high literacy attainment, including writing [3,108,109,110,111,112,113,114,115,116,117,118,119,120,121,122,123,124,125,126,127,128,129,130,131,132]. It also correlates positively to a facility with spoken language [133,134]. Sign language knowledge simply supports spoken language knowledge across the board [135,136,137], and this advantage goes both ways: sign language and spoken language promote the development of the other [67,127,133,138,139,140,141]. One way that signing can help a deaf child develop a spoken language is that signing directs visual attention to the participants in a conversation, an important step in communication in general and in recognizing lip actions [142] (see p. 1117) and in turn-taking in conversation [143]. Learning such visual attention can help with speechreading, in particular, which is what deaf children with CI rely on to varying degrees [144]. Indeed, sign language acquisition improves the linguistic and cognitive abilities of hearing children with developmental language disorders [145].

## 4. Understanding Bimodal-Bilingualism

Older views on the use of sign language and spoken language in the same environment misconstrue the nature of either modality and how each language is used by deaf people and deaf families as well as hearing families who successfully use sign language. The simple truth is that learning a language, spoken or sign, does not harm a child [146].

For example, Fitzpatrick and colleagues [104] offer what they call “a systematic review of the effectiveness of early sign and oral language intervention” and conclude that there is insufficient evidence that the addition of sign language fosters the acquisition of spoken language. That study is seriously flawed, however, as Hall [64] shows, since it does not distinguish between natural sign languages and artificial communication systems that use the hands—what are known as manually coded languages (MCLs, such as SEE mentioned earlier). This flaw is found in other works, as well, such as Geers and colleagues [147]. In fact, differences in findings between deaf children exposed to sign languages from deaf children exposed to other sign systems/MCLs are significant, where sign languages aid the deaf child in overall language development but MCLs do not [148]. MCLs lack characteristics of natural languages across the components of grammar. They are articulatorily unnatural in that they often “violate the perceptual fit between central and peripheral visual processes” [45] (p. 85). This makes MCLs difficult to use as a means of communication and inappropriate as a first language. While we do not know of studies addressing whether MCLs increase vocabulary, whatever positive effect they might have (or not have) in this regard is not reflected in the studies we have read. MCLs do not enable bilingualism or language transfer [119,149]. Using MCLs to test and measure the effectiveness of bimodal-bilingualism is misleading, because the bilingual component is missing, as MCLs are not naturally evolved languages. MCLs are an impoverished surrogate for either spoken language or sign language.

What do true bilingual-bimodal language interactions look like, then? In such environments sign language is used by as many people as possible in the child’s presence, whether the child is looking or not. In a natural environment, deaf children will respond to constant exposure to sign language. A spoken language is also present in the environment via speech, hearing, and print. For example, in the case of French and French Sign Language (LSF), French speech and hearing are also present in the environment as well as reading and writing. Both LSF and French are used in accordance with the deaf child’s ability to perceive and respond to them. This does not mean speaking and signing at the same time, but that combination does no harm as long as LSF is ever present and its structural integrity is not compromised by attempts to make LSF map directly onto French speech. It may be better to use LSF in some situations and French in different situations. Children growing up bilingual are not confused by using and switching between languages, even within the same sentence [150] (but see remarks in [151]), and as their overall linguistic competence grows, they successfully use mixed-language sentences just as adults do [152].

Bimodal-bilingualism for the family means using a natural sign language with the deaf infant or toddler throughout the day, and using a spoken language, as well, particularly reading and writing—its visual forms. If the child actively responds to speech, one would use a spoken language, as well as sign—not at the same time necessarily, but perhaps in tandem or alternatingly. Additionally, whether or not the child responds to speech, one would introduce the child to printed texts of the ambient spoken language—in shared reading activities at home, particularly with the help of recently developed bimodal-bilingual books [153,154,155,156]. These activities are most helpful when interactive strategies are employed, such as mimicry, asking open-ended questions, praising freely, and taking care to articulate signs in the child’s visual field [157]. In addition, doing structured activities targeting phonological awareness in a sign language as brief as 12 min daily for up to two months can produce positive effects on deaf children’s phonological awareness ([158,159]. An added benefit of the bimodal-bilingual approach is that the deaf child feels accepted and valued as a deaf person, which is a significant part of making a strong family bond [160,161].

## 5. Language Acquisition Begins at Home

Planning sign language acquisition in the home with appropriate family measures is happening in many countries, but this planning is largely undocumented and unresearched [162]. One difficulty is that institutional policies shift with each new player on the field exerting influence on the kinds and levels of support for families [163]. In the next section, we turn to the matter of language planning and policy in educational institutions. Here, instead, we focus on families who must cope with the confusing swirl of different recommendations.

Acquisition of sign language for the child begins with parents learning it as well, sometimes along with the child. As the parents’ fluency develops, so will the child’s access to language. Families and, particularly, caregivers need immediate and supportive information about sign language learning and connections to peer groups, especially since the early advice they receive substantially influences their attitudes toward deafness, thus, shaping the language planning they adopt [164]. Parents must have opportunities to learn sign language in a way that will not intimidate them nor make them feel incompetent, both for family bonding and because the degree to which a parent feels empowered and confident in their language choice with their deaf child correlates to better language and cognitive development of the child [165,166,167,168,169]. In fact, in a study of Spanish-speaking hearing mothers of deaf children in North America, a mother became so comfortable with signing that she used it naturally even with hearing members of the family without realizing it [170]. Most parents are highly motivated and learn to sign to varying degrees [171], where even moderate fluency offers great benefits to the child [161]. In 2014, 23% of families in the USA reported signing with their deaf children regularly [172].

Deaf children’s sign language acquisition will likely outpace their parents’ due to a child’s early childhood enhanced capacity to acquire language. Adults often do not become highly fluent, but they can still be effective communicators in sign language, which is what their child needs. Their child will acquire sign language naturally, transcending their parents’ abilities and being entirely fluent [173].

Different supports for helping families provide rich sign language input have been suggested. For example, Humphries and colleagues [174] call for government resources to fund sign language instruction for families of deaf children until the child is at least age 12. They also call for research on adult second-language learning in a second modality so that countries can improve the effectiveness of sign language instruction for hearing adults. Likewise, Koulidobrova, Kuntze, and Dostal [175] call for more resources to be allocated to parents who want to learn ASL and more research on how families can be helped in raising their child with sign as well as speech. Alfano [170] suggests parent-oriented workshops to teach parents how to communicate with their children, family get-togethers so that parents with deaf children can share strategies and resources on language activities, game-playing activities to promote ASL learning, and classes in ASL with transportation provided. Humphries and colleagues [2] urge parents to use all available resources, including doctors, local and national deaf community centers, schools for the deaf, deaf education services, articles, books, and the internet to find information and support. Weaver and Starner [176] offer a mobile app (SMARTSign) for learning ASL on one’s phone. Lillo-Martin, Gale, and Chen Pichler [177] outline, in detail, many actions that can help, from informing parents properly about bilingualism to informing medical professionals about language development in deaf children.

We know little about how well these suggestions work or what will optimize parental learning of sign [178], but we do know the importance of deaf people—individuals and communities—in the process of supporting deaf children to learn to sign [179,180,181,182,183,184,185,186] and in supporting families to learn how to help deaf children succeed [187,188,189,190,191] and to learn about deaf culture and deaf ways of managing in a hearing world [192]. Wille, Van Lierde, and Van Herreweghe [193] report that in Flanders, Belgium, where Flemish Sign Language was recognized as an official language in 2006, a parental support intervention practice was established for hearing parents of deaf children in 2014. The communicative strategies used by hearing parents and other family members with their deaf children varied from those used by deaf parents with deaf children (who all developed fluency in sign language). For example, hearing parents, in this study, relied more strongly on oral/aural strategies and explicit means of gaining visual attention (such as repositioning the child and waving), while deaf parents use more tactile and implicit means of gaining visual attention (such as waiting for the child to look up before signing). This case study demonstrates that hearing parents would benefit from learning the strategies used by deaf adults and deaf parents.

We agree with calls for research to find effective instruction for parents and for resources to pay for that instruction. In addition, we suggest an approach (as one best-practice example) that builds on the Utah State University Deaf Mentor Experimental Project described by Watkins, Pittman, and Walden [140]: deaf mentors made regular home visits to families with deaf children, ages 0–5 years, sharing ASL with them. In their three-year study comparing deaf children in the Utah Deaf Mentor program with deaf children without deaf mentors in Tennessee, where half the Tennessee children used only spoken language and the other half used total communication and MCLs, the children with deaf mentors exhibited greater linguistic, cognitive, and social progress than the children without deaf mentors. A study of a similar program involving deaf mentors in the UK found that children and families with mentors were more confident in the possibility of a good future for their children and this allowed families to set high expectations for the children and for children to have the confidence to strive to meet those expectations [194].

The approach we suggest here is also influenced heavily by evidence from a model used at the Seattle Children’s Hospital. There, Crezee, and Roat [195] found that bilingual patient navigators are more effective than professional interpreters in the success of the hospital’s treatment of deaf children as measured by “no-show rates, number of unplanned hospitalizations, average length of stay, and staff/family confidence in the family ability to care for the patient at home.” This success happened through a trust relationship, where a navigator helps everyone understand missed inferences, restates everything in plain language, alerts everyone to barriers to “implementation of treatment plans”, teaches basic skills in how to talk with teachers, and so on. Deaf mentors have, in fact, been a resource for decades [196], with efforts such as these implemented in varying places with varying consistency, and many models are possible [197,198].

Consider the following scenario. What if, upon the birth of a deaf child, a sign language “navigator” was made available to the family immediately? This person would be a bimodal-bilingual deaf adult trained to be a navigator. These individuals would:

be a bilingual model for the child as well as all family members;

help the family choose and sign up for sign language classes or tutoring;

be with the family for a significant number of hours each day using sign language with the child and all family members, helping them to communicate in various ways, including through signing, writing, and even gesturing;

help the family to connect to the local deaf community for socialization activities;

help the family to connect to other families with hearing parents and deaf children, as well as deaf families with deaf children;

help parents become advocates for their children [199].

The navigator can quell the family’s initial anxiety about their ability to raise a deaf child [200] and be there as a support at every step. As a result, bilingual and bimodal language interaction will be part of daily routines and play, which means everyone can exercise their right to enjoy the presence of a new baby in the house, promoting the well-being of the whole family [201] and everyone will have a consistent sign language model [202]. Further, parents will not be shy to communicate with their children in several modes, including pointing and other gestures. Mixed communication modes are natural in raising a hearing child [203,204,205,206]; there is no need to fear the same in raising a deaf child. Human communication is essentially multimodal, whether the primary language is speech or sign [207]. Parents can embellish their signing and speech with gestures they find natural as their own vocabulary in the sign language grows, knowing that the child always has a good signing model in the navigator.

This presence of a deaf adult can ease the family’s initial fears for their child’s future; the family will know a professional deaf adult well and be reassured that their own child can grow up to be a productive member of society. A deaf model, being bimodal-bilingual, can demonstrate to the family how people who are multilingual choose the language of a particular interaction [208,209]. Further, bilingualism can mean biculturalism, which has implications for behavior and identity [61].

Ideally, navigators would be available for a family from the birth of the child up to the point when the child enters primary school. In this way, the family will benefit from the presence of someone who knows the situation and all the relevant parties, and can help the family and medical and educational professionals in making recommendations that the family is able and likely to follow [198,210]. This approach requires that navigator training be available and that the government pay for deaf adults to work with families of deaf children. Then, a deaf adult can provide training in language and cognitive development milestones and can alert the relevant parties when needed.

The benefits to individuals and to society of such an approach to supporting parents will outweigh the costs by a significant amount. The human and economic benefits of investing in support for parents (e.g., in the form of navigator training) are well worth the cost, as deaf children will be far more likely to become happy, healthy, and productive members of society, contributing to the overall economic base of society. Hamilton and Clark [211] report that, as of 2020, a curriculum built around a deaf mentor and developed by Sensory Kids Impaired Home Intervention (SKI-HI) [212] is used in seven to ten states. Their study of families that made use of this particular deaf mentor program found that the children benefited by having a language-rich environment and by acquiring deaf practices of learning strategies and resources; they conclude that deaf mentors help secure the keys to success for deaf children.

## 6. Continuing Bimodal-Bilingualism into Schooling

Deaf children with hearing parents who are raised in a bimodal-bilingual environment can succeed in both languages [56], where success correlates positively with the length of time the child is enrolled in a bimodal-bilingual school environment [213], and where success in accessing a spoken language is strictly via its written form [117,214,215]. Neither the medical profession nor families are trained educators and, thus, they are not fully aware of what happens once the child goes to school. However, both the medical profession and parents can lobby for what should happen. It is important, then, to know what the bilingual-bimodal school environment looks like when practiced appropriately.

Howerton-Fox and Falk [216] argue that deaf children should be considered “English learners” and receive the kind of support given in the educational system to children for whom English is not their first language. Some countries have compared a bilingual approach to an oral-only approach for deaf children—with mixed results—but it is not always clear what their label of “bilingual” means. Razalli and colleagues [217], however, did a study with precise information. They compared the literacy achievements of deaf children in Malay using two educational programs. In one, spoken/written Malay was paired with Manually Coded Malay; in the other, spoken/written Malay was paired with Malay Sign Language. Deaf children achieved significantly higher skills in the program that used Malay Sign Language—that is, in a truly bilingual-bimodal program that compared two natural languages. To evaluate one type of educational approach vis a vis another, care must be taken in the research design to ensure that the sign language being evaluated is, indeed, a sign language and not a code for the spoken language.

Humphries and colleagues [2] argue that schooling for deaf children should have a focus on translanguaging, which involves everyone in the classroom using one’s full linguistic repertoire during activities and conversations. While the main components of translanguaging are the languages in the students’ and teachers’ repertoires—the sign languages and the spoken languages—many other forms of media are involved, such as print, video, recordings, and visuals. Deaf children coming to school from a home that is bilingual-bimodal will already have a head start in the translanguaging process. In the most supportive type of schooling, deaf children will be brought together in the classroom [218], so they can support each other socially and emotionally, as well as academically.

Another focus of schooling should be multi-literacy [219,220,221], including the preliteracy skills of understanding what a story is, what characters are, and why characters behave as they do in the stories. Vocabulary should be enhanced by building on sign language skills. There should be explicit teaching of phonemic awareness and morphological awareness in both spoken language and sign language texts [158,159]. Deaf children often benefit from lessons with a deaf teacher on the structure of their sign language, with explicit comparisons between the child’s sign language and the ambient spoken language [222] and on deaf cultural history and processes [223].

Humphries and colleagues [2] also make several schooling recommendations regarding inclusivity. Schooling is most effective when deaf children are grouped together in their grade level and assigned to teachers who have strong skills in signing. Being with other deaf students is critical to developing social skills and belonging in young children and youth. Qualified educational sign language interpreters should be in classrooms when there are staff and students who are emerging sign language learners [224]. Inclusivity is different from inclusion. Inclusion is often interpreted in ways that isolate deaf students from each other and from the language support they need. Bilingual bimodal schooling for deaf children makes sure deaf children are schooled in a rich academic, social, and cultural learning community.

## Data Availability

This study is not based on a corpus of data. Rather, the arguments come from the cited works.

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
