# Peer review of "Deaf Children Need Rich Language Input from the Start: Support in Advising Parents"

_children, 2022, doi:10.3390/children9111609_

Round 1

Reviewer 1 Report

This article is very well written and it mentions clear evidence that supports the authors' claims. I have only a few slight requests in order to better clarify some points:

When discussing the differences between MCLs and SL, the authors should also discuss the results from https://doi.org/10.1542/peds.2016-3489, as well as the explanations offered by https://doi.org/10.1093/oxfordhb/9780190054045.013.12 about possible biases in studies that claim that children exposed to oral communication have better outcomes than children exposed to SL.

With respect to the advantages of (sign language / spoken-written language) bilingualism over monolingualism (OC only) there is clear evidence also in https://doi.org/10.1093/deafed/ent045, https://doi.org/10.1017/S0022215112001909, and in https://doi.org/10.1017/S1366728913000849

Author Response

For reviewer 1, thank you!

Reviewer 3 also asked whether MCLs develop vocabulary, since, if they do, that would be something positive about them.  However, the studies we have read (and we have searched and searched) do not address whether MCLs increase vocabulary.  We now say that (lines 214-216 on page 5).

Reviewer 3 also asked us to talk not just about parents, but about caregivers and siblings.  We are entirely in accord with this.  Many of our cited articles, however, talk about parents.  So we now address this issue in the introduction and explain that we'll use the term "parent" as a cover term for all the relevant players.

Reviewer 3 also asked us to give more information on deaf mentoring programs.  This was a fabulous request.  We have now filled out Section 5 considerably.  It is far stronger now.  Thank you so much. 

Overall, the paper is greatly improved.  And the references are increased.  The reviewers were extremely helpful.

Reviewer 2 Report

I am grateful to have had the opportunity to review this paper. The authors tackle a difficult but necessary issue of forcefully advocating for sign language for families of deaf children. They begin by outlining the issue of being a deaf individual in a hearing environment, particularly a deaf children in hearing families, and especially linguistic, educational, and social impacts of language deprivation on a deaf child. They also discuss the issues surrounding manually-coded visual languages (MCLs) and cochlear implant (CI) use, and debunk myths surrounding access to sign language inhibiting later spoken language development. Finally they conclude with detailed and research-based suggestions for local- and country-wide support for hearing families with deaf children to achieve optimal success. 

I have no additional questions or comments and look forward to seeing this paper in press. 

Author Response

For reviewer 2, we had already cited the Davidson et al. 2014 paper and the Hassanzadeh 2012 paper.  But we now also cite the Geers et al. 2017 paper, the Rinaldi et al. 2020 paper, and the Rinaldi and Caselli 2014 paper.  Thank you.  These additions strengthen the work.  We are very grateful.

Reviewer 3 Report

The topic of this opinion piece/editorial is highly important as well as topical and its publication should be encouraged. However, it should be substantially strengthened with a number of changes/additions specifically to provide more substantial backing for the various recommendations made and also explicitly addressing and correcting (on the basis of research evidence) unsupported claims about the 'claimed negative effects of the use of a sign language on brain development and the acquisition of a spoken language. Without addressing the current argumentation against the early introduction of sign language, and providing counter-evidence within the text of the paper itself, it is likely to have limited impact on those it is most seeking to influence. The most important of these points include e.g.: more statistics on the percentage of those deaf children with cochlear implants who do not achieve (cognitively/educationally/socially) at an equivalent level to hearing peers, even in the context of good acquisition of spoken language. In relation to this, the importance of language, communication, and interaction assessment (in both spoken and signed language) should also be addressed. For example, a number of studies have shown that vocabulary size, whether in spoken or signed language is an important predictor of the successful development of literacy. A finding such as this provides reassurance that even grammatically limited language models may have a substantial impact. Much is made of a potential role for a "deaf navigator" but these arguments should be backed by more reference to findings of existing studies in which language and cultural support has been provided by deaf advisors- or at least some proposed research design which would provide key evidence if this is absent.

The paper also focusses primarily on parental sign language skills, although young deaf children grow up in many kinds of families, and the roles of other caregivers and siblings, in particular, could be emphasized more.

Author Response

For reviewer 3, you wanted us to give statistics regarding the efficacy of CIs.  It would be wonderful if we could do that.  it would be wonderful, in fact, if we could compare all deaf children in various language contexts with regard to the efficacy of the ways they are raised.  But clean comparisons are not possible.  Studies lump together different environments under the label "sign" and different environments under the label "spoken language only".  Thus we maintain our mode of arguing for early sign language exposure, and we give a brief explanation of that in our introduction now.

Reviewer 3 also asked whether MCLs develop vocabulary, since, if they do, that would be something positive about them.  However, the studies we have read (and we have searched and searched) do not address whether MCLs increase vocabulary.  We now say that (lines 214-216 on page 5).

Reviewer 3 also asked us to talk not just about parents, but about caregivers and siblings.  We are entirely in accord with this.  Many of our cited articles, however, talk about parents.  So we now address this issue in the introduction and explain that we'll use the term "parent" as a cover term for all the relevant players.

Reviewer 3 also asked us to give more information on deaf mentoring programs.  This was a fabulous request.  We have now filled out Section 5 considerably.  It is far stronger now.  Thank you so much. 

Overall, the paper is greatly improved.  And the references are increased.  The reviewers were extremely helpful.